# Ethnobotanical, Phytochemistry, and Pharmacological Activity of *Onosma* (Boraginaceae): An Updated Review

**DOI:** 10.3390/molecules27248687

**Published:** 2022-12-08

**Authors:** Ahmed Aj. Jabbar, Fuad O. Abdullah, Abdullah Othman Hassan, Yaseen Galali, Rawaz Rizgar Hassan, Essa Q. Rashid, Musher Ismael Salih, Kareem Fattah Aziz

**Affiliations:** 1Department of Medical Laboratory Technology, Erbil Technical Health and Medical College, Erbil Polytechnic University, Erbil 44001, Iraq; 2Department of Chemistry, College of Science, Salahaddin University-Erbil, Erbil 44001, Iraq; 3Department of Pharmacognosy, Faculty of Pharmacy, Tishk International University, Erbil 44001, Iraq; 4Department of Medical Laboratory Science, College of Science, Knowledge University, Kirkuk Road, Erbil 44001, Iraq; 5Department of Nutrition and Dietetics, Cihan University-Erbil, Erbil 44001, Iraq; 6Food Technology Department, College of Agricultural Engineering Sciences, Salahaddin University-Erbil, Erbil 44001, Iraq; 7Chemistry Laboratory Department, Kurdistan Medicine Control Agency (KMCA), Erbil 44001, Iraq; 8Department of Chemistry, Faculty of Science and Health, Koya University, Koya KOY45, Iraq; 9Department of Nursing, College of Nursing, Hawler Medical University, Erbil 44001, Iraq; 10Department of nursing, Faculty of Nursing, Tishk International University, Erbil 44001, Iraq

**Keywords:** *Onosma*, ethnobotany, phytochemistry, data mining, pharmacological activity, toxicology

## Abstract

The genus *Onosma* belongs to the *Boraginaceae* family and contains over 230 species. The present review sheds light on the ethnopharmacology, phytoconstituents, bioactivity, and toxicology of the *Onosma* species from previous investigations. Furthermore, the paper also highlights the unresolved issues for the future investigations. The review included previous studies of the genus *Onosma* available from Google Scholar and Baidu Scholar, Science Direct, SciFinder, Wiley Online Library, and Web of Science. Until now, more than 200 chemical compounds have been detected from the genus *Onosma*, including naphthoquinone (33), flavonoids (30), hydrocarbon (23), phenolic (22), ester (17), alkaloids (20), aromatics (12), carboxylic acid (11), fatty acids (9), terpenoids (10), while the most important ones are rosmarinic, ferulic, protocatechuic, chlorogenic, caffeic, p-coumaric acids, and apigenin. The *Onosma* species are reported as traditional medicine for wound healing, heart disease, and kidney disorders, while the pharmacological investigations revealed that the extracts and the phytochemicals of *Onosma* species have different therapeutic properties including antioxidant, enzyme inhibitory, antitumor, hepatoprotective, antiviral, anti-inflammatory, and antimicrobial actions. The summarized knowledge in this review provides valuable ideas for the current and future drug discovery and a motivation for further investigation on the genus *Onosma.*

## 1. Introduction

The genus *Onosma* comprises more than 230 species across the globe. The Asian continent has the highest share in terms of *Onosma* species existence [1], most of which are represented in Turkey by 88 species [2] followed by countries such as Iran and China by 58 [3] and 29 species [4], respectively. Iraqi Kurdistan represents 32 species of the genus *Onosma* based on the latest botanical studies [5]. However, recent investigations have revealed seven new species of *Onosma* in Asian countries, particularly Iran [6]. Continuous exploration on the ethnobotanical and plant taxonomy studies led to the discovery of several new *Onosma* species across our continent [7,8]. Some *Onosma* species have been well studied pharmacologically than others, and the most common ones are shown in Figure 1.

The ethnobotanical and in vitro studies have revealed that most of this *Onosma* species has many medicinal capabilities such as sedatives [9], antioxidant [10], anti-inflammatory [11], gastric disorders [11], antithrombotic [12], wound healing [13], Alzheimer [14], enzyme inhibitory [15], anti-tumor [16], anti-viral [17], antifungal [18], and COVID-19 curatives [19] (Figure 2).

The phytochemical studies on the genus *Onosma* have reported several chemical compounds as their main active ingredients, including naphthaquinones (5,8-dihydroxy-2-(4-methylpent-3-enyl) naphthalene-1,4-dione) [16], Phenolics (ferulic acid, vanillic acid), flavonoids (apigenin, luteolin) [10,20], alkannin, and shikonines (deoxyshikonin, isobutyrylshikonin, α-methylbutyrylshikonin, acetylshikonin) [21,22].

The past decades have showed numerous new records, phytochemical, and pharmacological studies of the new *Onosma* species, and the published two reviews were found lacking integrity as they contained incoherent data with skipping of some biological activities of the genus *Onosma* [23,24]. Therefore, in order to provide theoretical reference for further research and to comprehensively understand the medicinal applications of this genus, this article systematically reviewed traditional uses, chemical constituents, pharmacological activities, and clinical applications of the *Onosma* species based on the published literature.

## 2. Methodology

The authors independently extracted systematic literature data search from seven electronic databases: Google Scholar, PubMed, Science Direct, Sci-Finder, Wiley Online Library, Web of Science, and Baidu Scholar. The scientific name “*Onosma*” was searched to cover all relevant information from April 1800–2022, including folkloric uses, phytochemical contents, and pharmacological potentials (antimicrobial, anti-inflammatory, anticancer, antioxidant, enzyme inhibitory, and antidiabetic) of the *Onosma* species are presented in this review. More than 1000 articles were detected with keyword *Onosma*, about 132 articles were found with keyword *Onosma* phytochemical, 187 articles were found with keyword *Onosma* pharmacology, biological activity, medicinal uses, pharmacology, and toxicology. Out of these, 125 articles were published detailing the isolation and properties of different phytochemical contents of the *Onosma* genus, and a total of 95 papers were selected based on the quality, specificity, and the procedure of the investigation of *Onosma* extracts and its isolated compounds.

## 3. Regional (Folkloric) Name

The folkloric names of most *Onosma* species in most Middle east countries is Gaozaban, an Urdu word. It was first referred to *O. bracteatum* Wall, and among Arabic populations it is known as “Lisan-al-Thawr” or “Saqil ul-Hammam”. Furthermore, its English popular name is “Vipers Bugloss”, while, in Hindi language, the *Onosma* species known as “Ratanjot” as first referred to *O. echioides* L. [25,26].

## 4. Regional Distribution

Distribution map of some *Onosma* species collected from different regions of Iran, Iraq, and Turkey is shown in Figure 3. The point inputs to the models developed in this study were collected from their habitats of Iran such as Fars, Lorestan, Khuzestan, Kermanshah, Hamedan, Markazi, Ilam, Kohgiluyeh, Kerman, and Boyer-Ahmad provinces, and their habitats of Iraq include mainly some areas of Kurdistan, Sulaimani, Hawraman, Rwanduz, and Amedia districts. Meanwhile, their habitats of Turkey includes Sirt, Hakary, Anatolia, and Van [27,28,29].

## 5. *Onosma* Taxonomy

*Onosma* species belongs to the *Boraginaceae* family. The *Boraginaceae* family contains more than 100 genera and over 200 species, which are classified into five subfamilies: *Boraginoideae*, *Cordioideae*, *Ehretioideae*, *Hydrophylloideae*, and *Lennooideae* [30,31] (Table 1).

## 6. Traditional Use

The folkloric use of many *Onosma* species as medicinal plants for different health problems by local ethnic groups in several countries such as Iraq, Turkey, Iran, China, and India roots back to hundreds of years ago. Almost all plant parts, such as leaves, roots, underground parts, flowers, and the whole plant of this genus species are reported to have a broad range of therapeutic potentials (Table 2) [32,33]. Species such as *Onosma alborosea* have traditionally been utilized by Iraqi Kurdistan populations as a remedy for sedative, heart diseases, and kidney disorders through ingesting its aerial part extracts prepared by aqueous extraction methods [33]. The aerial parts of *Onosma orientalis* has been macerated with hot water for treating sedatives by Kurdish nations living in Iraqi Kurdistan [33]. Furthermore, the *O. armeniacum* K. has been used as Turkish folkloric medicine for healing wounds, peptic ulcers, burns, dyspnea, hoarseness, hemorrhoids, and abdominal pains through methods of cooking and filtration of its roots with butter [34]. The extracts (oil and aqueous extracts) of *O. argentatum* and *O. chlorotricum* has been traditionally utilized in Turkey and Iran (Lorestan province) for the treatment of wounds and cutaneous injures [35,36]. Furthermore, the root extracts of *O. hispidum* Wall. have been used traditionally by the Iranian nation as curatives for headache, wounds, insect stings, bits, and inflammatory diseases, while its flowers have been ingested for cardiovascular problems [37]. Moreover, same species has been used as a dye and as a substitute for alkanet [38]. The *O. bracteatum* Wall. extracts have been reported as traditional herbal medicine as a tonic agent for improving the body’s immune system with enhancing regulation of urine output [39]. The *O. bracteatum* Wall. also has been used as remedy for asthma, respiratory problems, tonic, alterative, demulcent, diuretic, spasmolytic, rheumatoid arthritis, diuretic, and antileprotic in India, Nepal, Kashmir, and northwestern Himalayas countries [40,41]. The root extracts of *O. sericeum* have been traditionally used in cream preparations for skin injuries and burn scar treatments in Adıyaman, Turkey [13].

The *O. microcarpum* has traditional medicine record for the healing of wounds and burn scars by rural residents of Il’yca district, Erzurum, Turkey [36,42]. The leaf aqueous extracts of *O. echioides* DC. are prepared for children suffering from constipation and metabolic disorders. Meanwhile, its flowers are reported as a cordial and as a stimulant for orthopedic and cardiac problems [43]. The dried roots of *O. paniculata* have a traditional medicinal record in Chinese herbal medicine for curing several human diseases including tumors [44]. The *O. aucheriana* is another species with traditional medicinal usage for itchiness, leucoderma, bronchitis, abdominal pain, strangury, fever, wounds, burns, and urinary calculi. Meanwhile, its flowers have been highlighted as stimulants and cardio-tonics, and its leaf extracts have been ingested as laxatives, purgatives, and as wound curatives [45]. Out of more than 230 species of *Onosma*, only 12 species were reported in traditional medicines as herbal medicine until now. This could be due to the large geographical distribution of the *Onosma* species and lack of scientific interest in the past, but this number is expected to increase in upcoming years as the researchers extensively search and investigate for other *Onosma* species after discovering some interesting phytochemical and pharmacological potentials of this genus in recent years.

**Table 2 molecules-27-08687-t002:** The traditional use of *Onosma* medicinal plants.

Species	Traditional Name	Country of Habitat	Medicinal Parts	Medicinal Use
*O. alborosea*		Safeen mountain, Shaqlawa district, Iraqi Kurdistan	Aerial parts	Sedative, heart diseases, kidney disorders [33]
*O. orientalis*		Safeen mountain, Shaqlawa district, Iraqi Kurdistan	Aerial parts	Sedative [33]
*O. armeniacum*		Turkey, Anatolia	Leaves	healing wound, peptic ulcers, burns, dyspnea, hoarseness, hemorrhoids, and abdominal pains [34]
*O. argentatum*,		Turkey	roots	Wound healing [35]
*O. chlorotricum*		Iran, Lorestan	roots	Wound healing [36]
*O. hispidum*		Iran (Korrassan)	roots	headache, wounds, insect stings and bits, inflammatory diseases, while its flowers are used for cardiovascular problems [37] and as a dye and a substitute for alkanet [38]
*O. bracteatum* Wall	Gaozaban, Sedge	India, Nepal, Kashmir, and in the northwestern Himalayas	Roots, flowers	asthma, respiratory problems, tonic, alterative, demulcent, diuretic, spasmolytic, rheumatoid arthritis, diuretic, and antileprotic [40,41].
*O. sericeum*		Turkey, Adıyaman	roots	As curatives for cutaneous wounds and burns [13]
*O. microcarpum*		Turkey, Il’yca district, Erzurum province	Roots and leaves	Wound healing [36]
*O. echioides*		Turkey	Leaves and flowers	Laxatives for children and as a cordial, stimulant for orthopedic and cardiac problems [46]
*O. paniculata*		China	roots	Anticancer [44]
*O. aucheriana*		Turkey	Roots, leaves, flowers	itchiness, leucoderma, bronchitis, abdominal pain, strangury, fever, wounds, burns, and urinary calculi. Stimulants and cardio-tonics. Laxative, purgative, and as wound remedy [45]

The traditional names, country, ingested parts, and medicinal purposes of the genus *Onosma* are listed in Table 2.

## 7. Chemical Profile of *Onosma* Species

The current systematic review of the phytochemical contents of *Onosma* species presents major identified organic classes such as naphthoquinone (33), flavonoids (30), hydrocarbon (23), phenolic (22), ester (17), alkaloids (20), terpenoids (10), carboxylic acid (11), fatty acids (9), aromatics (12), and liganin (5) compounds as shown in Figure 4. In addition, miscellaneous chemicals such as 24,25-Dihydroxycholecalciferol, 5-hydroxymethyl-furoic acid, and uplandicine also enrich the diversity of the phytochemistry in *Onosma* plants. Segregation of phytochemical contents in different classes is challenging and not always a clear and easy task. According to the current search, a total of 198 compounds are detected in the *Onosma* species as detailed in this review (Table 3), and this will open up new future study opportunities to explore pharmacological potentials of those phytochemicals. Most common *Onosma* compounds reported were rosmarinic acid, apigenin, ferulic acid, protocatechuic acid, chlorogenic acid, caffeic acid, p-coumaric acid, vanillic acid, luteolin, hyperoside, hesperidin, apigenin-7-Glucoside, luteolin-7, glucoside, isovalerylshikonin, acetylshikonin, pinoresinol, deoxyshikonin, 4, hydroxybenzoic acid, β,β-dimethylacryl, isovalerylshikonin, 2,5-Dihydroxybenzoic, and 3-Hydroxybenzoic acid as presented in Figure 5.

The bioactive structures of identified and characterized representative compounds, which are based on the repetition across published studies are shown in Figure 5, in addition to Figure 6.

## 8. Toxicity Study of the *Onosma* Species

### 8.1. Toxicity In Vivo Experiment

The chloroform and ethanolic extracts of *O. aucheranum*, *O. isauricum O. sericeum*, *O. tauricum*, and *O. tauricum* were safe in the administered doses from 100 mg/kg to 200 mg/kg based on the assessment of acute toxicity in the carrageenan-induced paw edema experiment as no abnormality in the morbidity nor mortality was recorded after 24 hours post treatment [73]. Furthermore, the 100, 200, 300, and 600 mg/kg of the MeOH of *O. mutabilis* administration to rats showed no changes in the appearance, behavior, and feed intake of the rats in a 7-day experiment [16]. Moreover, by the tarsal toxicity test, researchers have shown the acaricidal activity of the root extracts of *O. visianii* experimented against *Tetranychus urticae* mites in bean plants (*P. vulgaris* var. Carmen) after 24 h (considered as acute toxicity), which caused significant mortality of *T. urticae* adults with lethal doses 83.2 and 112.6 μg·cm causing 50% (LD_50_) and 90% (LD_90_) inhibition of oviposition, respectively. However, at 5 days (considered as chronic toxicity) from the start of the test, the lethal dose LD_50_ was more than 30 times lower (2.6 μg·cm^−2^) as a function of time used in the LD_50_ calculation [60]. Over the last two decades, several *Onosma* species have been tested for their toxicity to laboratory animal models. A study on toxicity of the bark extracts of *O. echioides* roots to Sprague Dawley rats (140 ± 10 g body weight) was performed and reported significant improvement in the body weight, food consumption, water intake, serum glucose, hematology, and biochemistry of rats with no adverse effect at a fixed dose [74].

### 8.2. Genotoxicity and Mutagenicity

Through the Allium-test, significant genotoxic effect from aqueous extracts of *O. stellulata* roots and aerial parts were observed in mitosis at meristematic cells of onion. Although the aerial parts showed significant genotoxicity after 4-h treatment (mitotic index was 2, 79%, vs. 9, 18% for control), but the root aqueous extracts had higher genotoxic effects. Genotoxic effects included changes in the structure of chromosomes (conglutination, spirality), and cytotoxic reaction and certain differentiation in the cell cycle, which were found to be in correlation with duration of treatment and solution concentration [75]. A genotoxic study by Allium anaphase–telophase assay reported that the safety of the ethanolic extract of *O. aucheriana* aerial parts at lower dose (62.5 mg/mL) had no toxic or genotoxic effects, while the higher dose (500 mg/mL) showed significantly the highest genotoxic effect including chromosomal aberrations, cells with multipolarity, cell bridges, and vagrant chromosomes (24.4%), cell fragments, and mitosis entrance [76]. In vivo genotoxic study of methanolic extracts of *O. sericea* and *O. stenoloba* at different doses (25, 50, 100, 200, and 400 μg/mL) against EMS-induced DNA damage in the flies and larvae of the wild-type strain of Drosophila melanogaster showed the absence of genotoxic effect of *O. sericea* and *O. stenoloba* at concentration 80 mg/mL. Furthermore, significant antigenotoxic effects reported after dual treatment with 80 mg/mL of both plant extracts plus EMS (ethyl methane sulfonate) caused significant decrease in DNA damage (with over 80% reduction) [15]. By using Ames assay, the antimutagenic potential of ethanolic extract of *O. bracteatum* has been reported against sodium azide and 2-aminofluorene mutagenicity in *Salmonella typhimurium* in TA100 strain (-S9 mix) as it displayed significant inhibition rate (82.30% at 250 mg/0.1 mL/plate), showing strong modulation of genotoxicity of base-pair substitution mutagen sodium azide when compared to NPD (frameshift mutagen) in TA98 tester strain. The *O. bracteatum* extracts showed significant antimutagenicity activity for preincubation mode than in co-incubation approach without -S9 in both TA100 and TA98 [77].

## 9. Pharmacological Activity of the *Onosma* Species

### 9.1. Antibacterial Activity

The essential oils isolated from roots of *O. sieheana* showed appreciable antibacterial activity against gram negative bacteria (*Escherichia coli* (MIC: 125 μg/mL) and *Pseudomonas aeruginosa* (MIC: 125 μg/mL) and gram positive bacteria (*Staphylococcus aureus* (MIC: 125 μg/mL) *and Bacillus subtilis* (MIC: 250 μg/mL)) [78]. The n-hexane–dichloromethane mixture extracts of *O. argentatum* roots showed antibacterial activity against *Bacillus subtilis*, *Escherichia coli*, and *Staphylococcus aureus* with MIC values 28, 13, and 32 μg/mL, respectively [18]. The chloroform fraction of *O. khyberianum* whole plant parts showed significant antibacterial activity against *Salmonella typhi*, *Shigella dysenteriae*, and Vibrio cholera inhibition zone 28, 26, 26 mm, respectively. Ethanol fraction of *O. khyberianum* demonstrated significant antiradical activity against *Shigella dysenteriae* (21 mm) and Vibrio cholera (20 mm), while the least active fraction of *O. khyberianum* n-hexane showed activity against *Vibrio cholera*, *S. aureus*, and *Shigella dysenteriae* (inhibition zone: 12, 9, 8 mm, respectively) but completely inactive against Salmonella and *E. coli* [79]. The crude ethanolic extracts of *O. hispidum* roots showed significant antibacterial activity against several gram positive and gram negative bacteria (*Corynebacterium diphtheria*, C.* diphtheriticum*, *Micrococcus lysodiecticus*, *S. aureus*, *S. epidermidis*, *S. saprophyticus*, *Enterococcus faecalis*, *E. faecalis 2400*, *E. faecium*, *Streptococcus pneumonia*, and *S. pyogenes*) with inhibition zone range between 18–20 mm [52]. The isolated naphtshoquinones (deoxyshikonin, isobutyrylshikonin, α- methylbutyrylshikonin, acetylshikonin, β-hydroxyisovalerylshikonin, 5,8-*O*-dimethyl isobutyrylshikonin, and 5,8-*O*-dimethyl deoxyshikonin) from *O. visanii* roots showed significant antibacterial activity against gram negative bacteria (*Citrobacter koseri*, *Hafnia alvei*, *maltophilia*, *Yersinia intermedia*, *Ps. proteolytica*, and *Stenotrophomonas*) and gram positive bacteria (*Bacillus megaterium*, *Enterococcus faecalis*, *S. epidermidis*, *Microbacterium arborescens*, and *Micrococcus luteus*) with MIC_50_ and MIC_90_ values between range 4.27–68.27 μg/mL and 4.77–76.20 μg/mL, respectively [61]. The antibacterial activity (MIC values) from methanol extract of aerial parts of *O. sericea* and *O. stenoloba* were between 2.5−10 mg/mL. Both *Onosma* extracts had moderate antibacterial activity only on a few strains, namely *A. chroococcum* and *E. coli* with MIC values 2.5 and 5 mg/L, respectively. *O. sericea* extract exhibited low activity on gram positive strain *M. lysodeikticus* with MIC 10 mg/mL, while *O. stenoloba* extract showed notable antibacterial action on *E. faecalis* and *A. tumefaciens* with MIC values 5 and 10 mg/mL, respectively [15].

### 9.2. Antifungal Activity

Antifungal activity of methanolic extracts of *O. sericea* and *O. stenoloba* aerial parts against fungal strains *Phialophare fastigiata and Fusarium oxysporum* has been reported as 2.5 and 5 μg/mLof MIC, respectively. Furthermore, the methanol extracts of *O. sericea* exhibited moderate activity (MIC range of 2.5−5 μg/mL) on *Penicillium canescens* FSB 24 and *P. cyclopium* FSB 23, while *O. stenoloba* had antifungal activity only against *P. cyclopium* (MIC 10 μg/mL). Moreover, the same study showed antifungal potentials (MIC 10 μg/mL) of *O. sericea* against *Trichoderma longibrachiatum* FSB 13 and *Trichoderma harzianum* FSB 12. Meanwhile, increased concentration (10 μg/mL) of *Onosma* extracts showed inactivity against *Aspergillus niger* FSB 31, *Aspergillus glaucus* FSB 32, *Doratomyces stemonitis* FSB 41, *Phialophora fastigiata* FSB 81, *Alternaria alternata* FSB 51, and *Fusarium oxysporum* FSB 91 [15]. The methanol extracts from aerial parts of *O. griffithii* exhibit antifungal activity against *Aspergillus flavus* (55%) and *Fusarium solani* (40%). Meanwhile, the chloroformic extracts showed better antifungal activity against *A. flavus* (59%) and *Fusarium solani* (60%) [17]. The antifungal activity of *O. kheberianum* against three fungal strains, *Fusarium oxysporum*, *Alternaria alternate*, and *A. flavus* were reported as 18, 13, and 7 mm, respectively, for ethanol fractions and 17, 11, and 9 mm, respectively, for chloroform fractions [79]. A previous study also showed a lack of antifungal activity of n-hexane–dichloromethane extracts of *O. argentatum* roots against *Trichophyton tonsurans*, *Trichophyton interdigitale*, *Microphyton gypseum*, and *Candida albicans* [18]. The essential oils from *O. sieheana* Hayek roots showed significant antifungal activity against yeast strains *Candida glabrata* and *C. albicans*, and the authors linked this activity with their phytoconstituents, namely Monoterpenes, such as cymene and thymol [80]. The essential oils from *O. chlorotricum* roots exhibit higher antifungal activity (21 and 19.3 mean of inhibition zones (mm) against *C. albicans* and *C. glaberata*, respectively) than that of essential oils from *O. microcarpum* roots [80]. The *O. paniculatum* cells showed strong response to fungal elicitors from Aspergillus sp., in an attempt to accelerate shikonin derivative formation and inversely arrest plant cell growth, which resulted in a slight change in shikonin contents [81].

### 9.3. Antioxidant Activity

*Onosma* species have been comprehensively studied and researchers have revealed that they are a promising resources of antioxidants using various types of extraction and solvent methods [14,82,83]. The *O. ambigens* aerial part extracts exhibited notable antioxidant action in the phosphomolybdenum, CUPRAC, FRAP, DPPH, and ABTS assays with values of 1.65, 0.95, 0.52, 1.86, and 1.45 mg/mL, respectively [30]. The antioxidant activity of *O. gigantea* were significant in phosphomolybdenum (134.31 μmol trolox (TEs)/g air dry matter (adm)), chelating effect (32.97 μmol (EDTAEs)/g adm), on DPPH (32.14 μmol TEs/g adm) and ABTS (58.68 μmol TEs/g adm)), and reducing power (CUPRAC (50.23 μmol TEs/g adm) and FRAP (40.96 μmol TEs/g adm)) assays [53].

The water extract of the aerial part of *O. pulchra* showed significant antioxidant actions in DPPH, ABTS, CUPRAC, and ferrous ion chelating tests (3.90, 2.55, 2.20, and 1.23 mg/mL, respectively). Meanwhile, the Phosphomolybdenum and FRAP assays showed superiority of MeOH extract (1.98 and 1.02 mg/mL, respectively) [14]. The ethanol extract of aerial parts of *O. bracteatum* showed significant radical quenching activity in superoxide radical scavenging (EC_50_: 115.14 μg/mL) and lipid peroxidation (EC_50_: 199.33 μg/mL) assays [77]. The methanol extracts of *O. mutabilis* showed higher antioxidant activity than that of water and ethyl acetate fractions, respectively, in which the antioxidant values for methanol extracts were 1.45 ± 0.05, 3.54 ± 0.064, 2.33 ± 0.045, 1.12 ± 0.023, and 1.62 ± 0.079 mg/mL in phosphomolybdenum, DPPH scavenging, ABTS, FRAP, and CUPRAC reducing, respectively [16]. The methanol extract of aerial parts of *O. frutescens* showed significantly higher antioxidant activity in DPPH (1.14 mg/mL), ABTS (1.04 mg/mL), CUPRAC (0.53 mg/mL), FRAP (0.35 mg/mL), and phosphomolybdenum (1.18 mg/mL) tests than that (1.75,1.50, 0.87, 0.55, 1.97 mg/mL) and (2.18, 1.87, 0.99, 0.63, 1.92 mg/mL) for *O. sericea* and *O. aucheriana*, respectively. The ferrous ion chelating assays showed superiority of O. aucheriana (IC_50_: 2.57 mg/mL) over *O. frutescens* (4.68 mg/mL) *and O. sericea* (6.18 mg/mL) [49]. The aqueous extract of O. aucheriana roots showed significant antioxidant activity in radical quenching activity (ABTS, DPPH) with IC50 values as 9.89 and 17.73 μg/mL. Additionally, the same species showed notable lipid peroxidation inhibition, and hydroxyl radical scavenging actions with IC50 values, 23.41 and 31.09 μg/mL, respectively [45]. The methanolic extracts of *O. trapezuntea* aerial parts showed stronger antioxidant activity (IC_50_: 3.05 mg/mL in DPPH and 7.19 mg/mL in ABTS) than that (IC_50_: 2.63 mg/mL in DPPH and 5.23 mg/mL in ABTS) of *O. rigidum* [50]. The *O. argentatum* root extracts (0.1% concentration) by n-hexane–dichloromethane mixture (1:1) showed significant 98% antioxidant activity (IC_50_: 0.0076% *w*/*v*) by thiobarbituric acid (TBA) [18]. The methanol extract of aerial parts of *O. lycaonica* Hub. -Mor. exhibited stronger antioxidant activity in 1,1-diphenyl-2-picrylhydrazyl scavenging activity (2.69 ± 0.10 mg/mL), cupric reducing antioxidant power (1.10 ± 0.01), ferric reducing antioxidant power (0.69 ± 0.01 mg/mL), and ferrous ion chelating activity (2.32 ± 0.16 mg/mL) than that of *O. papillosa*. However, the *O. papillosa* showed lower IC_50_ or *EC_50_* values for phosphomolybdenum (1.90 ± 0.07 mg/mL) when compared to *O. lycaonica* (2.05 ± 0.07 mg/mL), which could be related to their phytochemical contents as *O. lycaonica* had higher phenolic contents, with (43.5 ± 1.5 mg (gallic acid equivalent)/g extracts), whereas *O. papillosa* was higher in flavonoids (32.9 ± 0.3 mg (quercetin equivalent)/g extracts) [48]. The aerial part ethanol extracts of *O. hookeri* showed the same 2,2-diphenyl-1-picrylhydrazyl (77.77 ± 1.44 μg/mL) scavenging activity as butylated hydroxy toluene (72.70 ± 1.04 μg/mL), but slightly weaker 2,2′-azino-bis-3-ethylbenzthiazoline-6-sulphonic acid (553.56 ± 2.78 μg/mL) scavenging activity and total antioxidant capacity than that of BHT (51.44 ± 1.37 μg/mL), while the ethyl acetate fraction of *O. hookeri* showed better ABTS scavenger, with IC_50_ value of 84.83 ± 1.37 μg/mL [66]. The aerial part MeOH extracts of *O. sericea* significant antioxidant activity in DPPH scavenging (130.23 ± 5.31 mg TE/g extract), ABTS scavenging (235.53 ± 4.62 mg TE/g extract), FRAP (215.65 ± 2.51 mg TE/g extract), CUPRAC (359.63 ± 14.83 mg TE/g extract), total antioxidant capacity (2.46 ± 0.35 mmol TE/g extract), metal chelating activity (24.65 ± 2.21 mgEDTAE/g extract), while *O. stenoloba* stronger activity with values 53.96 ± 0.78, 95.60 ± 2.30, 76.48 ± 3.26, 142.88 ± 1.49 mg TE/g, 1.16 ± 0.05 mmol TE/g, and 5.51 ± 0.81 mg EDTAE/g in the same essays, respectively [81]. The aerial part extract of *O. isauricum* exhibited significant antioxidant actions with superiority of its methanol extracts in DPPH (34.75 mg/mL) and CUPRAC (0.643 mg/mL), ferric reducing powers (0.211 mg/mL), ABTS (188.68 mgTE/g extract), superoxide radical scavenging ability (97.50 mgTE/g extract), and total antioxidant ability (86.02 mgAAE/g extract) than that (31.44 mg/mL, 0.471 mg/mL, 0.237 mg/mL, 130.91 mgTE/g, 159.92 mgTE/g, 55.36 mgAAE/g) and (4.69 mg/mL, 0.078 mg/mL, 0.021 mg/mL, 131.94 mgTE/g, 103.23 mgTE/g, 31.17 mgAAE/g extract) for water and ethyl extracts, respectively [83]. The results of antioxidant investigations of *O. mollis* showed significant radical scavenging actions phosphomolybdenum, DPPH, and ABTS, (2.01, 3.33, 2.30 mg/mL, respectively) while reducing power activity, CUPRAC and FRAP, were found as 1.48 and 0.79 mg/mL, respectively [51].

### 9.4. Cytotoxicity Activity

For the past decades, several studies have confirmed the traditional usage of the *Onosma* species as cytotoxic agents, and mammalian cancer cell division was inhibited by its extracts and isolated compounds [45,55,60].

The methanol extract of *O. mutabilis* aerial parts indicated significant anticancer activity against prostate (DU-145), mammary (MCF-7), and cervical cancer (Hep2c) cells with IC_50_ values as 35.67 ± 0.15, 28.79 ± 0.23, and 41.83 ± 0.21 μg/mL, respectively [55]. The crude extracts of *O. aucheriana* showed significant cytotoxicity activity against human rhabdomyosarcoma, human cervix carcinoma Hep2c, and from murine fibroblast (L2OB) cell lines with IC_50_ values range between 25.54 to 50.57 μg/mL [45]. The isolated compounds acetylshikonin, dimethylacrylshikonin, α-methylbutyrylshikonin, and isovalerylshikonin from the roots of *O. paniculata* showed appreciable anticancer activity against human CCRF-CEM leukemia, MDA-MB-231 breast cancer, human U251 glioblastoma, HCT 116 colon cancer, and human melanoma (SBcl2, WM35, WM9, WM164) cell lines with IC_50_ values ranging between 600 nM to 70 μM [60]. The isolated naphtshoquinones α-methylbutyrylshikonin and acetylshikonin compounds from *O. visanii* roots demonstrated stronger cytotoxic activity against MDAMB-231 cells (IC_50_: 86.0 μg/mL and 80.2 μg/mL, respectively) than that of 118.9, 204.6, 424.7, 391.6, and 411.5 μg/mL of Deoxyshikonin, β-Hydroxyisovalerylshikonin, Isobutyrylshikonin, 5,8-*O*-Dimethyl deoxyshikonin, and 5,8-*O*-Dimethyl isobutyrylshikonin, respectively. Additionally, all compounds except 5,8-*O*-Dimethyl deoxyshikonin, and 5,8-*O*-Dimethyl isobutyrylshikonin reduced viability of MDA-MB-231 cells after 48 h of incubation. Furthermore, α-methylbutyrylshikonin demonstrated the higher anticancer activity against HCT116 cells (IC_50_: 15.2 μg/mL) than that 97.8 μg/mL, 24.6 μg/mL and 30.9 μg/mL of Deoxyshikonin, Acetylshikonin, and β-Hydroxyisovalerylshikonin, respectively [61]. The effect of *Onosma bracteatum* has been studied against different cancer cell lines and the results showed that various concentrations (0.055, 0.11, 0.22, 0.44, 0.88, 1.7, and 3.52 µg/mL) of *O. bracteatum* decreased viability of cells in a time- and dose-dependent protocol [84]. Furthermore, the hydrochloric root extracts of *O. dichroanthum* Boiss. roots have shown significant anticancer actions against gastric cancer cells [11]. Moreover, *O. paniculata* has shown notable cytotoxicity activity against a number of cancer lines and linked their action with its ability to accelerate apoptosis [60]. The 50 µg/mL ethanolic extract from aerial parts of *O. sericeum* exhibited significant cytotoxicity activity against the breast cancer cells (MCF-7) with significantly decreased cell viability (28.76 ± 11.31%) [13]. The petroleum ether and aqueous extracts of *O. hispidum* roots have shown significant anticancer actions against HepG2 liver cancer cell lines [85].

### 9.5. Enzyme Inhibitory Activity

#### 9.5.1. Antidiabetic Activity

A literature search revealed multiple research works that confirmed the anti-diabetics properties of *Onosma* species as the in vitro antidiabetic activity of *Onosma* species was reported based on its inhibitory potentials on α-amylase and glucosidase enzymes. The ethyl acetate extraction of aerial parts of *O. gigantea* showed higher α-amylase and glucosidase inhibitory activity (15.98 and 1.07 μmol/g) than that (410.50 and 6.75 μmol/g) and (1320.53 and 5.16μmol/g) of methanol and water extracts, respectively [53]. The α-amylase inhibitory activity from MeOH extracts of *O. aucheriana* and *O. sericea* were reported higher (2.50 and 2.51 mg/mL, respectively) than that (3.15 mg/mL) of *O. frutescens* [49]. The ethyl acetate extraction of *O. ambigens* aerial parts showed stronger α-amylase inhibitory activity (IC_50_: 2.64 mg/mL) than that (2.98 and 16.34 mg/mL) for methanol and water extracts, respectively [30]. The methanol extracts of *O. lycaonica* and *O. papillosa* aerial parts exhibited significant α-amylase inhibitory concentration (IC_50_: 2.57 and 2.40 mg/mL) and glucosidase inhibition (IC_50_: 2.60 and 2.61 mg/mL), respectively [48]. The ethyl acetate extract of *O. pulchra* aerial parts showed higher α-amylase inhibitory activity (2.40 mg/mL) than that (5.47 and 19.23 mg/mL) of methanol and water extracts, respectively [14]. The aerial part extraction of *O. rigidum* showed higher glucosidase and lower α-amylase enzyme inhibitory activity than that of *O. trapezuntea* extracts [50]. The MeOH aerial extracts of *O. stenoloba* exhibited higher α-amylase and lower glucosidase inhibitory activity (0.89 and 43.47 mmol/g) than that (1.26 and 33.38 mmol/g) of *O. sericea*, respectively [15]. The hydroalcoholic extract of the aerial part of *O. Dichroanthum* was reported to have anti-diabetic and anti-neuropathy properties based on its ability to down regulation of the MDA and Glutathione levels in homogenized tissues of brain and liver in a rat experiment [86]. The petroleum ether, chloroform, and methanol extracts of *O. hispidum* wall roots have shown significant anticancer actions with inhibitory percentages reported as 70, 58, and 50%, respectively. Meanwhile, the superiority of petroleum ether extracts has been linked with its higher polyphenolic contents [85].

#### 9.5.2. Alzheimer’s Disease

The protective effect of *Onosma* species against Alzheimer’s disease was reported depending on its inhibitory activity on acetylcholinesterase (AChE) and butyrylcholinesterase (BChE) enzymes. The ethyl acetate extraction of aerial parts of *O. gigantea* showed higher AChE and BChE inhibitory activity (2.76 and 6.87 μmol/g, respectively) than that (31.57 and 1.82 μmol/g, respectively) of methanol extracts [53]. The isolated hispidone and (2S)-5,2-dihydroxy-7,5-dimethoxyflavanone from methanol extractions of whole plant parts of *O. hispida* showed significant inhibitory activity against AChE (11.6 and 15.7 mg/mL, respectively) and BChE (28.0 and 7.9 mg/mL, respectively) enzymes [38]. The aerial part extracts of O. *lycaonica* and *O. papillosa* exhibited significant AChE inhibition activity (IC_50_:1.32 and increased BChE inhibitory activity (2.31 ± 0.04 and 2.07 ± 0.1 (2.31 ± 0.04 and 2.07 ± 0.08 mg GALAEs/g extracts), respectively [48]. The MeOH extraction of *O. rigidum* aerial parts showed higher AChE and lower BChE inhibitory activity than that of *O. trapezuntea* extracts [50]. *O. sericea* aerial part extracts showed higher inhibitory activity on AChE (3.74 mg/g) and BChE (0.51 mg/g) than that (4.34 and 3.44 mg/g) for *O. stenoloba*, respectively [15].

#### 9.5.3. Anti-Tyrosinase Activity

Tyrosinase enzymes are well-known for their participation in melanin biosynthesis, and hypersecretion accompanied by accumulation of melanin pigments may lead to hyperpigmentation disorders and photo carcinogenesis [87]. The ethyl acetate partition of aerial parts of *O. gigantea* showed higher tyrosinase inhibitory activity (0.15 μmol/g) than that (0.49 and 10.48μmol/g) of methanol and water extracts, respectively [53]. The tyrosinase inhibitory activity of methanol extracts of *O. aucheriana* aerial parts was higher (2.19 mg/mL) than that (2.23 and 2.40 mg/mL) of *O. sericea* and *O. frutescence*, respectively [49]. The methanol partition of aerial parts of *O. ambigens* showed higher tyrosinase inhibitory activity (2.81 mg/mL) than that (3.79 and 4.45 mg/mL) of water and ethyl acetate extracts, respectively [30]. *Onosma lycaonica* and *O. papillosa* aerial extracts have been reported as tyrosinase inhibitors with IC_50_ values 2.20 and 2.05 mg/mL, respectively [48]. The methanol extracts of *O. pulchra* aerial parts showed higher tyrosinase inhibitory activity (2.47 mg/mL) than that (3.77 and 4.35 mg/mL) of ethyl acetate and water extracts, respectively [14]. The aerial part extracts of *O. rigidum* and *O. trapezuntea* showed comparable tyrosinase inhibitory potentials activity [50]. A previous study also reported modest tyrosinase inhibitory activity (136.35 and 135.68 mg/g) for methanol extracts of aerial parts of *O. sericea* and *O. stenoloba*, respectively [15]. The ethyl acetate extracts of *O. isauricum* showed higher tyrosinase inhibitory activity (19.96 mg/g kojic acid equivalents) than that (15.33 and 14.83 mg/g) of methanol and water extracts, respectively [83].

#### 9.5.4. Anti-Lipoxygenases Activity

Lipoxygenases enzymes are known to catalyze oxidation of polyunsaturated fatty acids (linoleic, linolenic, and arachidonic acid) yielding hydroperoxides. Such reactions may be favorable, but also lipoxygenases may interact undesirably. Aromatic compounds are major yields of lipoxygenase reactions that can interfere with food properties, mainly during long-term storage. Lipoxygenase’s impact on unsaturated fatty acids may lead to off-flavor/off-odor formation, leading to food spoilage. Furthermore, lipoxygenase is considered as an important enzyme in stimulation of inflammatory reactions in the human body by playing as a key factor in the biosynthesis of many bio-regulatory compounds such as hydroxyeicosatetraenoic acids (HETEs), leukotrienes, lipoxins, and hepoxylines that were linked to major diseases such as cancer, stroke, and heart and brain diseases [88]. Therefore, searching for natural products that could target this enzyme has become a continuous scientific mission to prevent such diseases. The onosmins A (2-[(4-methylbenzyl)amino]benzoic acid and B (methyl 2-[(4-methylbenzyl)amino]benzoate) compounds isolated from the n-hexane-soluble fraction of ethanol extracts of *O. hispida* whole plant showed significant lipoxygenase inhibitory activity (IC_50_: 24.0 and 36.2 μM) [54].

## 10. Other Biological Activity

### 10.1. Parasiticidal Activity

The antileishmanial activities of the crude methanol extract of *O. griffithii* and its fractions were statistically significant (*p* < 0.05) against the *Leishmania promastigotes*, Pakistani isolates in comparison with the standard drug called *Pentamidine* [17].

### 10.2. Anti-Inflammatory and Analgesic Activity

The chloroform extracts from roots of *O. aucheranum*, *O. isauricum*, and *O. tauricum* showed 28.0%, 34.3%, and 15.6% inhibitory action in p-benzoquinone-induced abdominal constriction experiment, while the ethanol extracts of *O. isauricum* and *O. sericeum* demonstrated inhibition action of 24.6% and 27.5%, respectively, in the same test. The chloroform and ethanol extracts of *O. isauricum* and ethanol extract of *O. sericeum* also showed significant inhibitory activity, ranging between 12.3–27.3%, 10.5–25.3%, 8.2–22.6%, respectively, in a carrageenan-induced hind paw edema model at 100 mg/kg dose without gastric damage, and the activity was very comparable to indomethacin (32.0–38.4% inhibition) as a standard sample [73]. The chloroform extracts of *O. aucheranum* and *O. isauricum* and ethanolic extracts of *O. isauricum* and *O. sericeum* exhibited notable antinociceptive activity; 28.0%, 34.3%, 24.6%, and 27.5% inhibition, respectively, against p-benzoquinone-induced abdominal contractions, without induction of any sign of gastric lesion [73]. The methanol extraction of aerial parts of *O. bracteatum* showed potent analgesic activity by inducing significant increase in the latency period in a dose-dependent manner at different doses at 1, 2, and 3 h (with superiority of 500 mg/kg i.e., 258.9% (*p* < 0.05) at 3 h) post feeding, respectively, in a tail flick test. Furthermore, the methanol extract of *O. bracteatum* showed significant analgesic effect at 500 mg/kg body weight dose by inducing 54% inhibition (*p* < 0.05) in comparison to 45.9% inhibition activity for standard Diclofenac sodium (5 mg/kg body weight) [89].

### 10.3. Gastric-Ulcerogenic Activity

The chloroform and ethanol extracts from *O. aucheranum*, *O. isauricum*, *O. sericeum*, and *O. tauricum* roots did not cause any gastric lesions or bleeding in the stomach of mice in a 48-h experiment [71].

### 10.4. Treatment and Prevention of COVID-19

The *Onosma* phytochemicals, deoxyshikonin, 3-hydroxy-isovaleryl shikonin, propionyl shikonin, and acetyl shikonin showed significant binding affinities for the Mpro enzyme based on the molecular docking studies using two distinct approaches, in which a SiteMap module of Maestro was used to detect the possible ligand binding sites for the Mpro enzyme. Docking simulations and molecular mechanics suggest that shikonin derivatives might be effective anti-SARS-CoV-2 compounds [19].

## 11. Conclusions

Application of natural products and their metabolites as chemically diverse starting building blocks has been a major driving force in drug discovery over the last century. However, the use of natural products is not linked only to the modern era, as most folkloric medicines have plant-derived extracts. Moreover, the technological advancement and new technical development for isolation and identification of the natural bioactive compounds in herbs have motivated scientists to investigate and use them as nutrients and nutraceuticals, as well as curatives.

The genus *Onosma*, known to be widespread worldwide, has a history of medicinal uses against different diseases in the folk medicine system of several civilizations. In this review, the authors rediscover the genus *Onosma* by detailing the important isolated and identified chemical compounds and extracts, including naphthoquinone (33), flavonoids (30), hydrocarbon (23), phenolic (22), ester (17), alkaloids (20), terpenoids (10), carboxylic acid (11), fatty acids (9), aromatics (12), and liganin (5). The *Onosma* phytoconstituents that are considered as potential leads amenable for drug development were reported as rosmarinic acid, apigenin, ferulic acid, protocatechuic acid, chlorogenic acid, caffeic acid, p-coumaric acid, vanillic acid.

Several biological activities were reported from *Onosma* compounds and extracts, including, *Genotoxicity and Mutagenicity*, antifungal, antibacterial, antioxidant, anticancer, antidiabetic, anti-Alzheimer, anti-tyrosinase, anti-lipoxygenases, parasiticidal, anti-inflammatory, and gastric-ulcerogenic activities. Finally, despite the fact that rosmarinic acid is reported as the most detectable compound in the *Onosma* species, it was not found in other species such as *O. echioides*, *O. hookeri*, *O. heterophylla*, and *O. erecta*, requiring further investigation for more confirmation by profiling many other species for comparison.

## Figures and Tables

**Figure 1 molecules-27-08687-f001:**
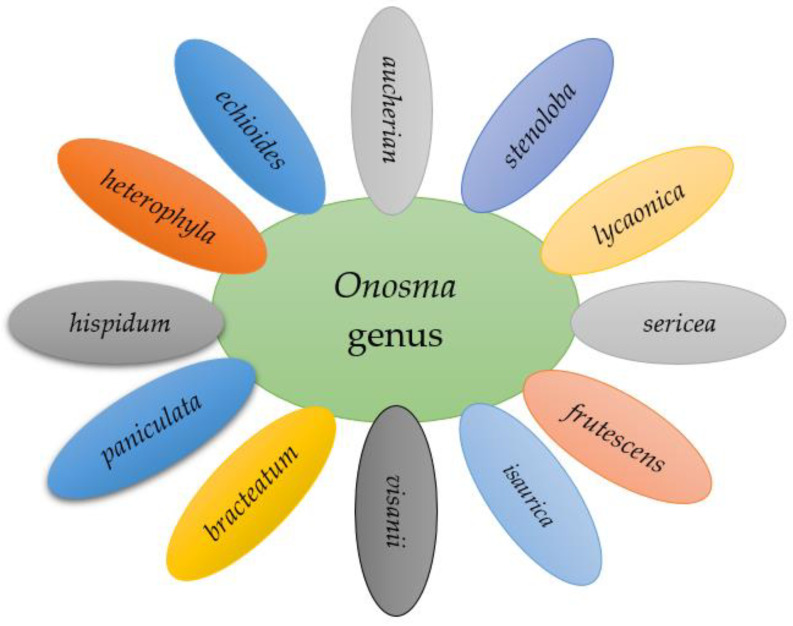
Representative example of the most-studied species of the genus *Onosma* in terms of pharmacology actions.

**Figure 2 molecules-27-08687-f002:**
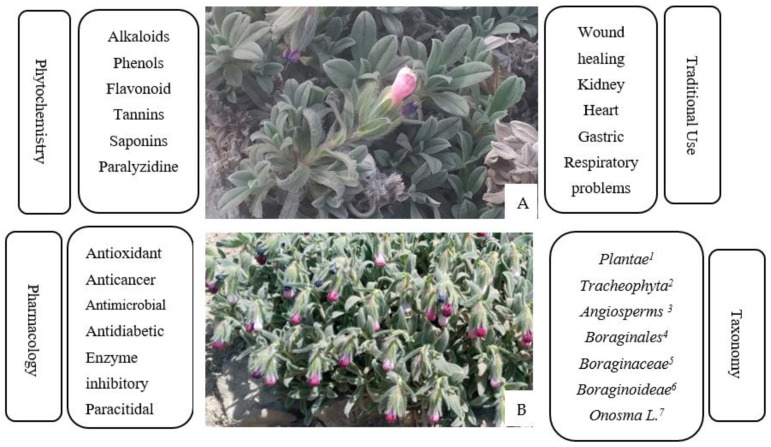
Chemical profile and pharmacological activities of the *Onosma species* ((**A**) *O. mutabilis*, (**B**): *O. Alborosea). 1*: Kingdom, *2*: Phylum, *3*: Class, *4*: Order, *5*: Family, *6*: Subfamily, *7*: Genus.

**Figure 3 molecules-27-08687-f003:**
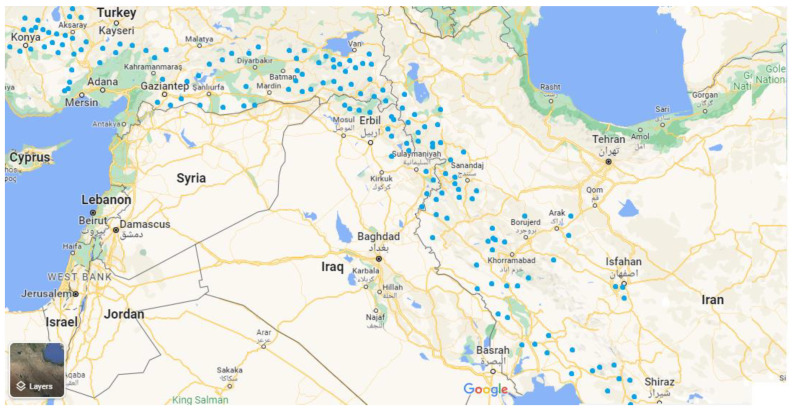
Regional distribution of some *Onosma* species in Iran, Iraq, and Turkey [4,27,29].

**Figure 4 molecules-27-08687-f004:**
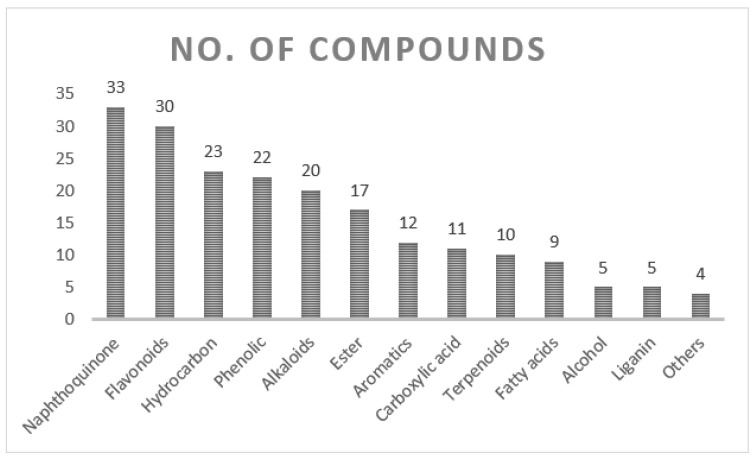
Organic class contents of *Onosma* species based on reported compounds.

**Figure 5 molecules-27-08687-f005:**
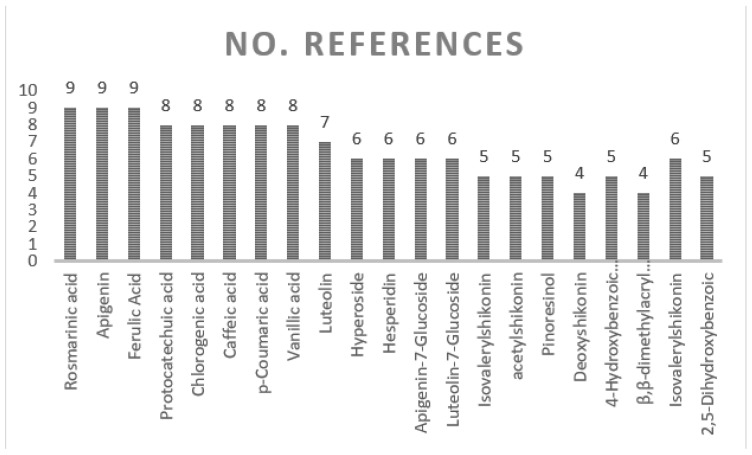
Most common *Onosma* compounds based on the repetition in the literature.

**Figure 6 molecules-27-08687-f006:**
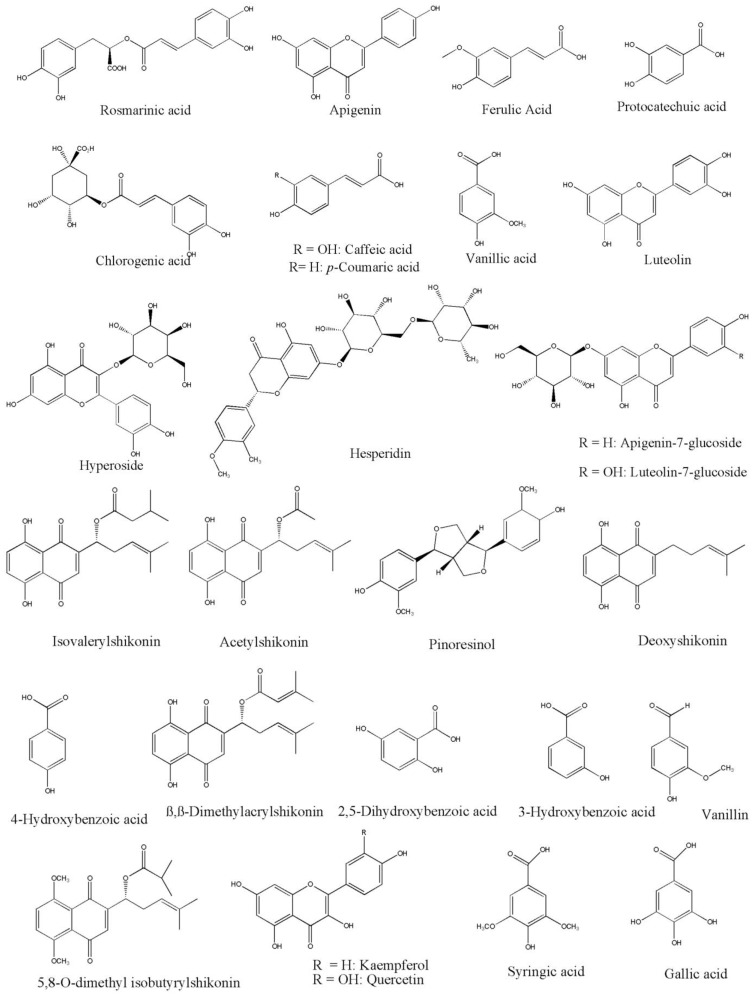
Representative of main compounds isolated from *Onosma* species, repeated in the literature.

**Table 1 molecules-27-08687-t001:** *Onosma* taxonomy according to Global Biodiversity Information Facility [1].

Kingdom	Plantea
Phylum	*Tracheophyta*
Class	*Angiosperms*
Order	*Boraginales*
Family	*Boraginaceae*
Subfamily	*Boraginoideae*
Genus	*Onosma* L.

**Table 3 molecules-27-08687-t003:** Names and sources of compounds isolated from genus *Onosma*.

No.	Chemical Names	Organic Class	Plant Species	Distribution in Plant	Reference	
**1**	Hyperoside	Flavonoid	*O. isaurica*, *O. bracteosa*, *O. lycaonica*, *O. papillosa*, *O. pulchra*, *O. frutescens*, *O. aucheriana*, *O. sericea*, *O. trapezuntea*, *O. rigidum*, *O. mollis*, *O. inexspectata*, *O. armenum*	Aerial parts	[10,14,47,48,49,50,51]	
**2**	Hesperidin	Flavonoid	*O. isaurica*, *O. bracteosa*, *O. lycaonica*, *O. papillosa*, *O. ambigens*, *O. pulchra*, *O. frutescens*, *O. aucheriana*, *O. sericea*, *O. mollis*, *O. inexspectata*, *O. armenum*	Aerial parts	[10,14,30,47,48,49,51]	
**3**	Vanillic acid	Aromatics	*O. isaurica*, *O. bracteosa*, *O. ambigens*, *O. pulchra*, *O. frutescens*, *O. aucheriana*, *O. sericea*, *O. bracteatum*, *O. inexspectata*, *O. armenum*, *O. hispidum*, *O. mollis*	Aerial parts	[10,14,30,39,47,49,51,52]	
**4**	Pinoresinol	Phenolics	*O. isaurica*, *O. bracteosa*, *O. ambigens*, *O. pulchra*, *O. frutescens*, *O. aucheriana*, *O. sericea*, *O. mollis*, *O. sericea*, *O. inexspectata*, *O. armenum*	Aerial parts	[10,14,30,47,49,51]	
**5**	Apigenin-7-glucoside	Flaconoid	*O. isaurica*, *O. bracteosa*, *O. lycaonica*, *O. papillosa*, *O. ambigens*, *O. pulchra*, *O. frutescens*, *O. aucheriana*, *O. sericea*, *O. mollis*, *O. inexspectata*, *O. armenum*	Aerial parts	[10,14,30,47,48,49,51]	
**6**	Apigenin	Flavonoid	*O. isaurica*, *O. bracteosa*, *O. lycaonica*, *O. papillosa*, *O. ambigens*, *O. gigantea*, *O. pulchra*, *O. frutescens*, *O. aucheriana*, *O. sericea*, *O. hispida*, *O. mollis*, *O. inexspectata*, *O. armenum*	Aerial parts	[10,14,30,47,48,49,51,53,54]	
**7**	Ferulic acid	Phenolics	*O. isaurica*, *O. bracteosa*, *O. sericea*, *O. lycaonica*, *O. papillosa. O. aucheriana*, *O. gigantea*, *O. pulchra O. frutescens*, *O. inexspectata*, *O. armenum*, *O. hispidum*, *O. mollis*	Aerial parts	[10,14,15,47,48,49,51,52,53,55]	
**8**	Luteolin-7-glucoside	Flavonoid	*O. isaurica*, *O. bracteosa*, *O. lycaonica*, *O. papillosa*, *O. ambigens*, *O. pulchra*, *O. frutescens*, *O. aucheriana*, *O. sericea*, *O. mollis*, *O. inexspectata*, *O. armenum*	Aerial parts	[10,14,30,47,48,49,51]	
**9**	Luteolin	Flavonoid	*O. isaurica*, *O. bracteosa*, *O. stenoloba*, *O. lycaonica*, *O. papillosa. O. gigantea*, *O. pulchra*, *O. frutescens*, *O. aucheriana*, *O. inexspectata*, *O. armenum*, *O. sericea*, *O. mollis*	Aerial parts	[10,14,15,47,48,49,51,53]	
**10**	Rosmarinic acid	Aromatic	*O. isaurica*, *O. inexspectata*, *O. armenum*, *O. bracteosa*, *O. lycaonica*, *O. papillosa*, *O. ambigens*, *O. aucheriana*, *O. gigantea*, *O. pulchra*, *O. frutescens*, *O. sericea*, *O. bracteatum*, *O. trapezuntea*, *O. rigidum*, *O. inexspectata*, *O. armenum*, *O. mutabilis*, *O. mollis*	Aerial parts	[10,14,16,30,39,45,47,48,50,51,53]	
**11**	3-Hydroxybenzoic acid	Carboxylic acid	*O. isaurica*, *O. bracteosa*, *O. pulchra*, *O. aucheriana*, *O. sericea*, *O. inexspectata*, *O. armenum*	Aerial parts	[10,14,47,49,51]	
**12**	Protocatechuic acid	Carboxylic acid	*O. isaurica*, *O. bracteosa*, *O. gigantea*, *O. pulchra*, *O. frutescens*, *O. aucheriana*, *O. sericea*, *O. ambigens*, *O. bracteatum*, *O. mollis*, *O. inexspectata*, *O. armenum*	Aerial parts	[14,30,39,45,47,49,51,53]	
**13**	Chlorogenic acid	Quinic acids	*O. isaurica*, *O. bracteosa*, *O. ambigens*, *O. aucheriana*, *O. gigantea*, *O. pulchra*, *O. frutescens*, *O. sericea*, *O. trapezuntea*, *O. rigidum*, *O. mollis*, *O. inexspectata*, *O. armenum*	Aerial parts	[14,30,45,47,49,50,51,53]	
**14**	Gentisic acid	Carboxylic acid	*O. isaurica*, *O. bracteosa*, *O. pulchra*, *O. frutescens*, *O. aucheriana*, *O. sericea*, *O. lycaonica*, *O. papillosa*, *O. mollis*	Aerial parts	[14,47,48,49]	
**15**	Caffeic acid	Carboxylic acid	*O. isaurica*, *O. bracteosa*, *O. lycaonica*, *O. papillosa. O. aucheriana*, *O. gigantea*, *O. pulchra*, *O. bracteatum*, *O. inexspectata*, *O. armenum*	Aerial parts	[10,14,39,45,47,48,51,53]	
**16**	p-Coumaric acid	Aromatics	*O. isaurica*, *O. bracteosa*, *O. aucheriana*, *O. gigantea*, *O. pulchra*, *O. frutescens*, *O. sericea*, *O. lycaonica*, *O. papillosa*, *O. ambigens*, *O. inexspectata*, *O. armenum*	Aerial parts	[10,14,30,45,46,47,48,49,51,53]	
**17**	Salvianic acid A	Phenolics	*O. stenoloba*, *O. sericea*	Aerial parts	[15]	
**18**	Verbascoside	Phenolics	*O. sericea*, *O. aucheriana*,	Aerial parts	[15,49]	
**19**	Rosmarinic acid-*O*-hexoside	Aromatics	*O. sericea*, *O. stenoloba*	Aerial parts	[15]	
**20**	Apigenin-*O*-hexoside	Flavonoid	*O. sericea*, *O. stenoloba*	Aerial parts	[15]	
**21**	Methyl caffeate	Phenolics	*O. sericea*, *O. stenoloba*	Aerial parts	[15]	
**22**	Apigenin-*O*-rhamnosylhexoside	Phenolics	*O. sericea*, *O. stenoloba*	Aerial parts	[15]	
**23**	Diosmin	Flavonoid	*O. sericea*	Aerial parts	[15]	
**24**	O-Methylrosmarinic acid isomer	Phenolics	*O. sericea*	Aerial parts	[15]	
**25**	Tricin	Flavonoid	*O. sericea*	Aerial parts	[15]	
**26**	Cirsiliol	Flavonoid	*O. sericea*	Aerial parts	[15]	
**27**	Diosmetin	Flavonoid	*O. sericea*	Aerial parts	[15,55]	
**28**	Stearic acid	Fatty acid	*O. sericea*	Aerial parts	[15]	
**29**	Intermedine	Ester	*O. stenoloba*, *O. alborosea*, *O. arenaria*	Aerial parts, roots	[15,56]	
**30**	Lycopsamine	Alkaloid	*O. stenoloba*	Aerial parts	[15]	
**31**	Caffeoylshikimic acid isomer	Phenolics	*O. stenoloba*	Aerial parts	[15]	
**32**	Heliosupine	Pyrrolizidine Alkaloids	*O. stenoloba*	Aerial parts	[15]	
**33**	Vicenin-2	Flavonoid	*O. stenoloba*	Aerial parts	[15]	
**34**	Echimidine	Pyrrolizidine Alkaloids	*O. stenoloba*	Aerial parts	[15]	
**35**	Isoferulic acid	Aromatics	*O. stenoloba*	Aerial parts	[15]	
**36**	Rosmarinic acid-di-Ohexoside	Aromatics	*O. stenoloba*	Aerial parts	[15]	
**37**	Quercetin-*O*-hexoside	Flavonoid	*O. stenoloba*, *O. sericea*	Aerial parts	[15]	
**38**	Kaempferol-*O*-hexoside	Ester	*O. stenoloba*, *O. sericea*	Aerial parts	[15]	
**39**	Isorhamnetin-*O*-rhamnosylhexoside	Flavonoid	*O. stenoloba*	Aerial parts	[15]	
**40**	Trihydroxyisoflavone	Flavonoid	*O. stenoloba*	Aerial parts	[15]	
**41**	Ursolic acid		*O. stenoloba*	Aerial parts	[15]	
**42**	4-Hydroxybenzoic acid	Triterpenoids	*O. lycaonica*, *O. papillosa*, *O. ambigens*, *O. pulchra*, *O. frutescens*, *O. aucheriana*, *O. sericea*, *O. gigantea*, *O. aucheriana*, *O. bracteatum*	Aerial parts	[14,30,39,48,49]	
**43**	Eriodictyol	Carboxylic acid	*O. lycaonica*, *O. papillosa.*	Aerial parts	[48]	
**44**	Vanillin	Phenolics	*O. lycaonica*, *O. papillosa. O. pulchra*, *O. frutescens*, *O.aucherian*, *O. sericea*	Aerial parts	[14,48,49,57]	
**45**	(+)-Catechin	Flavonoid	*O. lycaonica*, *O. papillosa O. frutescens*	Aerial parts	[48,49,57]	
**46**	Homoprotocatechuic acid	Phenolics	*O. lycaonica*, *O. papillosa*	Aerial parts	[48,57]	
**47**	Acetylshikonin	Naphthoquinones	*O. heterophylla*	Roots	[58]	
**48**	Shikonin derivatives	Naphthoquinones	*O. heterophylla*	Roots	[58]	
**49**	Acetyl shikonin	Naphthoquinones	*O. heterophylla*	Roots	[58]	
**50**	Shikonin derivatives	Naphthoquinones	*O. visianii*	Roots	[59]	
**51**	Shikonin derivatives	Naphthoquinones	*O. visianii*	Roots	[59]	
**52**	Isobutyrylshikonin	Naphthoquinones	*O. visianii*	Roots	[21,59]	
**53**	Isovalerylshikonin	Naphthoquinones	*O. visianii*, *O. paniculata*, *O. exsertum*, *O. waltonii*, *O. paniculatum*, *O. hookeri*, *O. confertum*, *O. echioides*, *O. heterophylla*	Roots	[21,22,59,60,61,62]	
**54**	α-methylbutyrylshikonin	Naphthoquinones	*O. visianii*	Roots	[21,59]	
**55**	5,8-*O*-dimethyl deoxyshikonin	Naphthoquinones	*O. visianii*	Roots	[59,63]	
**56**	5,8-*O*-dimethyl isobutyrylshikonin	Naphthoquinones	*O. visianii*	Roots	[21,59,61]	
**57**	deoxyshikonin	Naphthoquinones	*O. visianii*, *O. paniculata*, paniculatum	Roots	[21,59,60,62,63]	
**58**	Acetylshikonin	Naphthoquinones	*O. visianii*, *O. confertum*, *O. echioides*, *O. setosum*, *O. paniculata*, paniculatum	Roots	[21,59,60,61,64]	
**59**	β-Hydroxyisovalerylshikonin	Naphthoquinones	*O. paniculata*, *O. heterophylla*	Roots	[63,65]	
**60**	β,β-dimethylacrylshikonin	Naphthoquinones	*O. paniculata*, *O. confertum*, *O. exsertum*, *O. waltonii*, *O. paniculatum*, *hookeri*, *Onosma hookeri*, *Onosma zerizaminum*	Roots	[62,63,64,65]	
**61**	Methylbutyrylshikonin	Naphthoquinones	*O. paniculata*	Roots	[63]	
**62**	Isovalerylshikonin	Naphthoquinones	*O. paniculata*	Roots	[63]	
**63**	Gallic acid	Fatty acid	*O. aucheriana*, *O. pulchra*, *O. frutescens*, *O. sericea*	Aerial parts	[14,45,49]	
**64**	Quercetin	Flavonoid	*O. aucheriana*, *O. pulchra*, *O. frutescens*, *O. sericea*	Aerial parts	[14,45,49]	
**65**	Syringic acid	Fatty acid	*O. aucheriana*, *O. pulchra*, *O. frutescens*, *O. sericea*	Aerial parts	[14,49]	
**66**	Shikonin derivatives	Hydrocarbon	*O. mutabilis*	Aerial parts	[16]	
**67**	Shikonin derivatives	naphthoquinones	*O. mutabilis*	Aerial parts	[16]	
**68**	3-*O*-Methyl-d-glucose	Hydrocarbon	*O. mutabilis*	Aerial parts	[16]	
**69**	24,25-Dihydroxycholecalciferol	Vitamin D	*O. mutabilis*	Aerial parts	[16]	
**70**	β-Sitosterol	Phytosterol	*O. mutabilis*, *O. heterophylla*	Aerial parts, roots	[16,65]	
**71**	Phenol, 2,4-bis(1,1-dimethylethyl)-, phosphite	Phenolics	*O. mutabilis*	Aerial parts	[16]	
**72**	p-Hydroxybenzoic acid	Carboxylic acid	*O. gigantea*, *O. aucheriana*, *O. bracteatum*	Aerial parts	[39,45,53]	
**73**	trans-Cinnamic acid	Cinnamic acid	*O. gigantea*	Aerial parts	[53]	
**74**	Kaempferol	Flavonoid	*O. gigantea*, *O. pulchra*, *O. frutescens*, *O. aucheriana*, *O. sericea*	Aerial parts	[14,49,53]	
**75**	3,4-Dihydroxyphenylacetic acid	Catechol	*O. pulchra*	Aerial parts	[14]	
**76**	Taxifolin	Flavonoid	*O. pulchra*, *O. frutescens*, *O. aucheriana*, *O. sericea*	Aerial parts	[14,49]	
**77**	Sinapic acid	Aromatics	*O. pulchra*, *O. frutescens*, *O. aucheriana*, *O. sericea*	Aerial parts	[14,49]	
**78**	Eriodictyol	Flavonoid	*O. pulchra*, *O. frutescens*, *O. aucheriana*, *O. sericea*	Aerial parts	[14,49]	
**79**	Shikonin derivatives	Naphthoquinones	*O. echioides*	Aerial parts	[64]	
**80**	Pulmonarioside C	Phenolics	*O. bracteatum*	Aerial parts	[39]	
**81**	9′-Methoxyl salvianolic acid	Flavonoid	*O. bracteatum*	Aerial parts	[39]	
**82**	4-*O*-(E)-p-coumaroyl-l-threonic acid	Phenolics	*O. bracteatum*	Aerial parts	[39]	
**83**	Coumarin	Aromatics	*O. bracteatum*	Aerial parts	[39]	
**84**	Umbelliferone	Aromatics	*O. bracteatum*	Aerial parts	[39]	
**85**	Scopoletin	Aromatics	*F*	Aerial parts	[39]	
**86**	6,7-Dimethoxycoumarin	Aromatics	*O. bracteatum*	Aerial parts	[39]	
**87**	Esculetin	Aromatics	*O. bracteatum*	Aerial parts	[39]	
**88**	Caffeic acid methyl ester	Ester	*O. bracteatum*	Aerial parts	[39]	
**89**	1-*O*-Caffeoyl glycerol	Phenolics	*O. bracteatum*	Aerial parts	[39]	
**90**	Latifolicinin C	Phenolics	*O. bracteatum*	Aerial parts	[39]	
**91**	Oresbiusin A	Phenolics	*O. bracteatum*	Aerial parts	[39]	
**92**	Ethyl 3-(3, 4-dihydroxyphenyl)lactate	Phenolics	*O. bracteatum*	Aerial parts	[39]	
**93**	4, 5-Dihydroxy-3-methoxybenzoic acid	Carboxylic acid	*O. bracteatum*	Aerial parts	[39]	
**94**	5-Hydroxymethyl-furoic acid	Furoic acid	*O. bracteatum*	Aerial parts	[39]	
**95**	3,4-Dihydroxybenzyl alcohol	Alcohol	*O. bracteatum*	Aerial parts	[39]	
**96**	Rosmarinic acid methyl ester	Ester	*O. bracteatum*	Aerial parts	[39]	
**97**	Salviaflaside methyl ester	Ester	*O. bracteatum*	Aerial parts	[39]	
**98**	9′-(2,3-Dihydroxypropyl)-rosmarinic acid	Phenolics	*O. bracteatum*	Aerial parts	[39]	
**99**	p-Coumarinic acid ester of trigonotin	Ester	*O. bracteatum*	Aerial parts	[39]	
**100**	Echiumin A	Liganin	*O. bracteatum*	Aerial parts	[39]	
**101**	Ternifoliuslignan A	Liganin	*O. bracteatum*	Aerial parts	[39]	
**102**	Ternifoliuslignan D	Liganin	*O. bracteatum*	Aerial parts	[39]	
**103**	Eritrichin	Liganin	*O. bracteatum*	Aerial parts	[39]	
**104**	Shikonin derivatives	Naphthoquinon	*O. bracteatum*	Aerial parts	[39]	
**105**	Kaempferol 3-*O*-[α-l-rhamnopyranosyl-(1→2)-β-d-glucopyranoside]	Flavonoid	*O. bracteatum*	Aerial parts	[39]	
**106**	Kaempferol 3-*O*-[α-l-rhamno pyranosyl-(1→6)-β-d-glucopyranoside]	Flavonoid	*O. bracteatum*	Aerial parts	[39]	
**107**	Impecylone A	Flavonoid	*O. bracteatum*	Aerial parts	[39]	
**108**	Tigloylshikonin	Naphthoquinon	*O. hookeri*	Roots	[66]	
**109**	Acetyl shikonin	Naphthoquinon	*O. hispidum*	Roots	[67]	
**110**	Alkannan	Naphthoquinon	*O. hispidum*, *O. echioides*	Roots	[67]	
**111**	Deoxyshikonin	Naphthoquinon	*O. hispidum*, *O. echioides*, *O. confertum*	Roots	[62,67]	
**112**	7-*O*-acetylechinatine N-oxide	Alkaloid	*O. erects*	Roots	[68]	
**113**	Viridinatine N-oxide stereoisomer	Alkaloid	*O. erects*	Roots	[68]	
**114**	7-Epi-echimiplatine Noxide	Alkaloid	*O. erects*	Roots	[68]	
**115**	Onosmerectine N-oxide	Alkaloid	*O. erects*	Roots	[68]	
**116**	Acid 2,3-dimethyl-2,3,4-trihydroxypentanoic acid	Alkaloid	*O. erects*	Roots	[68]	
**117**	Acyloin 4-methyl-2-hydroxypentanon	Alkaloid	*O. erects*	Roots	[68]	
**118**	2-Methyl-n-butyrylshikonin	Naphthoquinon	*O. exsertum*, *O. waltonii*, *O. paniculatum*, *hookeri*, *O. confertum*	Roots	[62]	
**119**	β-Acetoxyisovalerylshikonin	Naphthoquinon	*O. exsertum*, *O. waltonii*, *O. paniculatum*, *O. hookeri*, *O. confertum*	Roots	[62]	
**120**	Isobutylshikonin	Naphthoquinon	*O. exsertum*, *O. waltonii*, *O. paniculatum*, *O. hookeri*, *O. confertum*	Roots	[62]	
**121**	Alkannin	Naphthoquinon	*O. echioides*, *O. paniculata*	Roots	[65]	
**122**	Shikonin	Naphthoquinon	*O. caucasicum*, *O. conferitum*, *O. hookeri*, *O. livanovii*, *O. polyphyllum*, *O. tauricum*, *O. sericium*, *O. setosum*, *O. visianii*, *O. zerizaminium*	Roots	[65]	
**123**	β,β-dimethylacrylalkannin	Naphthoquinon	*O. heterophylla*, *O. hookeri*, *O. paniculata*	Roots	[65]	
**124**	Heliotridine	Alkaloid	*O. heterophyllum*	Roots	[58]	
**125**	Necine derivative (1-methyl-8(-pyrrolizine)	Alkaloid	*O. heterophyllum*	Roots	[58]	
**126**	Acetylintermedine	Alkaloid	*O. alborosea*, *O. arenaria*	Roots	[56,57]	
**127**	O7-Acetyllycopsamine	Alkaloid	*O. alborosea*, *O. arenaria*	Roots	[56,57]	
**128**	5,6-Dihydro-7,9-dimethoxy 7H- pyrrolizine	Alkaloid	*O. arenaria*	Roots	[56]	
**129**	7-Acetylretronecine	Alkaloid	*O. arenaria*	Roots	[56]	
**130**	9-(Butyryl-2-ene) supinidine		*O. arenaria*	Roots	[56]	
**131**	7-Acetyl-9-(2-methylbutyryl) retronecine	Alkaloid	*O. arenaria*	Roots	[56]	
**132**	7-Acetyl-9-(2,3-dimethylbutyryl) retronecine	Alkaloid	*O. arenaria*	Roots	[56]	
**133**	7-Acetyl-9-(2-hydroxy-3-methylbutyryl) retronecine	Alkaloid	*O. arenaria*	Roots	[56]	
**134**	3′-Acetylsupinine	Alkaloid	*O. arenaria*	Roots	[56]	
**135**	7-Acetyl-9-(2,3-dihydroxybutyryl) retronecine	Alkaloid	*O. arenaria*	Roots	[56]	
**136**	Uplandicine	Pyrrolizines	*O. arenaria*	Roots	[56]	
**137**	Palmitic acid	Fatty acid	*O. irrigans*	Fruits	[69]	
**138**	Oleic acid	Fatty acid	*O. irrigans*	Fruits	[69]	
**139**	Linolenic acid	Fatty acid	O. irrigans	Fruits	[69]	
**140**	γ-Linolenic acid	Fatty acid	*O. irrigans*	Fruits	[69]	
**141**	Stearidonic acid	Fatty acid	*O. irrigans*	Fruits	[69]	
**142**	monoenoic acids 20:1, 22:1, and 24:1	Fatty acid	*O. irrigans*	Fruits	[69]	
**143**	Hexahydrofarnesyl acetone	Fatty acid	*O. bulbotrichum*, *O. isaurica*	Aerial parts	[70]	
**144**	Phytol	Diterpenoid	*O. bulbotrichum*, *O. isaurica*	Aerial parts	[70]	
**145**	Farnesyl acetone	Diterpenoid	*O. bulbotrichum*, *O. isaurica*	Aerial parts	[70]	
**146**	Hexadecanal	Hydrocarbon	*O. bulbotrichum*, *O. isaurica*	Aerial parts	[70]	
**147**	Hexyl hexanoate	Ester	*O. isaurica*	Aerial parts	[70]	
**148**	(E)-2-Decenal	Medium-chain aldehyde Hydrocarbon	*O. bulbotrichum*	Aerial parts	[70]	
**149**	1-Hexadecene	Unsaturated aliphatic Hydrocarbon	*O. isaurica*	Aerial parts	[70]	
**150**	Safranal	Hydrocarbon	*O. isaurica*	Aerial parts	[70]	
**151**	Heptadecane	Hydrocarbon	*O. bulbotrichum*	Aerial parts	[70]	
**152**	Dodecanal	Hydrocarbon	*O. bulbotrichum*, *O. isaurica*	Aerial parts	[70]	
**153**	E)-2-Undecenal	Hydrocarbon	*O. bulbotrichum*	Aerial parts	[70]	
**154**	Tridecanal	Hydrocarbon	*O. bulbotrichum*	Aerial parts	[70]	
**155**	(E)-Geranyl acetone	Diterpenoid	*O. bulbotrichum*, *O. isaurica*	Aerial parts	[70]	
**156**	1-Isobutyl-4-isopropyl-2,2-dimethyl succinate	Dicarboxylic acid	*O. bulbotrichum*	Aerial parts	[70]	
**157**	Neophytadiene isomer I	Terpenoid	*O. isaurica*	Aerial parts	[70]	
**158**	Tetradecanal	Hydrocarbon	*O. bulbotrichum*	Aerial parts	[70]	
**159**	(E)-β-Ionone	Sesquiterpenoid	*O. bulbotrichum*	Aerial parts	[70]	
**160**	Neophytadiene	Sesquiterpenoid	*O. isaurica*	Aerial parts	[70]	
**161**	Pentadecanal	Sydrocarbon	*O. bulbotrichum*	Aerial parts	[70]	
**162**	(E)-Nerolidol	Sesquiterpenoid	*O. bulbotrichum*	Aerial parts	[70]	
**163**	Hexadecanal	Hydrocarbon	*O. bulbotrichum*, *O. isaurica*	Aerial parts	[70]	
**164**	3,4-Dimethyl-5-pentylidene-2(5H)-furanone	Phenolics	*O. bulbotrichum*, *O. isaurica*	Aerial parts	[70]	
**165**	3,4-Dimethyl-5-pentyl-5H-furan-2-one	Phenolics	*O. bulbotrichum*, *O. isaurica*	Aerial parts	[70]	
**166**	Carvacrol	Monoterpenoid	*O. bulbotrichum*	Aerial parts	[70]	
**167**	Tricosane	Hydrocarbon	*O. bulbotrichum*, *O. isaurica*	Aerial parts	[70]	
**168**	(2E, 6E)-Farnesol	Sesquiterpenoid	*O. bulbotrichum*	Aerial parts	[70]	
**169**	Tetracosane	Hydrocarbon	*O. isaurica*	Aerial parts	[70]	
**170**	Pentacosane	Hydrocarbon	*O. bulbotrichum*, *O. isaurica*	Aerial parts	[70]	
**171**	Geranyl linalool	Hydrocarbon	*O. isaurica*	Aerial parts	[70]	
**172**	Heptacosane	Hydrocarbon	*O. bulbotrichum*, *O. isaurica*	Aerial parts	[70]	
**173**	Nonacosane	Hydrocarbon	*O. isaurica*	Aerial parts	[70]	
**174**	1-Docosene	Unsaturated aliphatic Hydrocarbon	*O. isaurica*	Aerial parts	[70]	
**175**	isorhamnetin-3-*O*-rutinoside	Flavonoid	*O. stellulata*	Aerial parts	[71]	
**176**	sinapic acid	Aromatic	*O. stellulata*	Aerial parts	[71]	
**177**	Deoxyshikonin [2-(4-methyl-pent-3-enyl)-5,8-dihydroxynaphthalene-1,4-dione]	Naphthoquinon	*O. nigricaule*	roots	[72]	
**178**	β, β - Dimethylacrylshikonin (5,8-Dihydroxy-2-[1-(β, β -dimethy lacryloyloxy)-4-methyl-3-pentenyl]-1,4-naphthalenedion]	Naphthoquinon	*O. nigricaule*	Roots	[72]	
**179**	Acetyl shikonin [(+)-Acetic acid 1-(5,8-dihydroxy-1,4- dioxo-1,4-dihydro-naphthalen-2-yl)-4-methyl-pent-3-enyl ester]	Naphthoquinon	*O. nigricaule*	Roots	[72]	
**180**	2-[(4-methylbenzyl)amino]benzoic acid	Carboxylic acid	*O. hispida*	Whole plant	[54]	
**181**	Methyl 2-[(4-methylbenzyl)amino]benzoate	Flavonoid	*O. hispida*	Whole plant	[54]	
**182**	6,4′-Dimethoxy-3,5,7-trihydroxyflavone	Flavonoid	*O. hispida*	Whole plant	[54]	
**183**	apigenin 7-*O*-β-d-glucoside	Flavonoid	*O. hispida*	Whole plant	[54]	
**184**	Paraffins	Hydrocarbon	*O. heterophylla*	Roots	[62]	
**185**	n-Dodecane	Hydrocarbon	*O. heterophylla*	Roots	[62]	
**186**	n-Decatrian	Hydrocarbon	*O. heterophylla*	Roots	[62]	
**187**	Methyl dodecanoate	Methyl ester	*O. heterophylla*	Roots	[62]	
**188**	Methyl tetradecanoate	Methyl ester	*O. heterophylla*	Roots	[62]	
**189**	Methyl 4-methyl tetradodecan-9,12 dien-oate	Hydrocarbon	*O. heterophylla*	Roots	[62]	
**190**	Methyl 4-methyl tetradodec-9-ene-oate	Hydrocarbon	*O. heterophylla*	Roots	[62]	
**191**	Methyl 4-methyl hexadec-9-ene-oate	Methyl ester	*O. heterophylla*	Roots	[62]	
**192**	Methyl hexadecanoate	Methyl ester	*O. heterophylla*	Roots	[62]	
**193**	Ethyl hexadecanoate	Methyl ester	*O. heterophylla*	Roots	[62]	
**194**	Isopropyl hexadecanoate	Methyl ester	*O. heterophylla*	Roots	[62]	
**195**	Methyl octadeca-9,12,15-triene-oate	Methyl ester	*O. heterophylla*	Roots	[62]	
**196**	Methyl octadeca-9,12-diene-oate	Methyl ester	*O. heterophylla*	Roots	[62]	
**197**	Methyl octadec-9-ene-oate	Methyl ester	*O. heterophylla*	Roots	[62]	
**198**	Methyl octadecanoate	Methyl ester	*O. heterophylla*	Roots	[65]	
**199**	diosmetin-7-*O*-β-glucoside	Aromatics	*O. bourgaei*	Aerial parts	[55]	
**200**	allantoin	Imidazoles	*O. bourgaei*	Aerial parts	[55]	
**201**	globoidnan A	Liganin	*O. bourgaei*	Aerial parts	[55]	

## Data Availability

Not applicable.

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
