# Peer review of "Ethnobotanical, Phytochemistry, and Pharmacological Activity of Onosma (Boraginaceae): An Updated Review"

_molecules, 2022, doi:10.3390/molecules27248687_

Round 1

Reviewer 1 Report

The manuscript, entitled “Ethnobotanical, Phytochemistry, and Pharmacological activity of Onosma (Boraginaceae): An updated review”, have summarized the study advance in ethnopharmacology, chemical components, bioactivities, and toxicology of Onosma. The review will help people and researchers to deeply know this genus and may benefit to the further study and comprehensive utilization of Onosma species. However, the current version of the manuscript is unsuitable to publish on the Journal for the reasons: 1) The figures are hard to read for the low resolution, redundant marks, and information absence; 2) The writing seems arbitrary, there are some misleading results and lots of format issues can be found in the manuscript; 3) Some sentences were confused and make difficulty in understanding, it need to reorganize and a bit of polishing. 

The detailed comments are:  

1.     Line 50, it is not correct to take Iran and China as south east countries.

2.     Line 56, “in vitro” should be italic. “as” in Line 57 should be “such as”.

3.   For the figures in the text, the wavy lines in Figure1 and Figure 2 should be removed; No species have been shown on the map of Turkey in Figure 3 while the figure title has mentioned Turkey; The exact number of compounds and references are need to attach on the each histogram in Figure 4 and Figure 5; The “Figure 5.1” should be “Figure 6”, and the structures and names of each compounds should be more clear and large for readable. The resolutions of Figure 1-3 are needed to improve.

4.     For the tables in the text, the colons in Table1 and dots behind numbers in Table 3 should be removed; Table 3 need to add a column named “Chemical classification” and all compounds need to reorganize following the chemical classification.

5.     Line122-123, the sentence is confused, underground parts should not belong to plant organs.

6.  Line 142, the citation “Badruddeen et al., 2012; Ved, et al., 2016” is not consistent with citation format, it should be changed into “[41, 42]”. The same issue also can be found in Line 275.

7.     Line 169-170, the classification of compounds is disordered. Many types in this sentence are overlapped, such as phenolic and flavonoids. The classification should be redone based on IUPAC rule or followed authoritative literatures and give the corresponding citation.

8.     The scientific name of species should be used the formats of italic and full name for the first appear. “O. sericea” in Line 276 and “O. stenoloba” in Line 280 need to replace to “O. sericea” and “O. stenoloba”; “P. fastigiata and F. oxysporum” in Line 284 should be changed into “Phialophare fastigiata and Fusarium oxysporum”.

9.   Line 328, the Unit of bioactivity values should be added. The same issues also present in Line 345-347, Line 379, Line 485, etc.

10.  Line 339-341, based on the given IC50 values, the authors have made a wrong result for the activity comparation.

11.  “mg/mL” or “mg/ml” should be only choice one and keep it in whole text.

12.  Line 410-412, the sentence is confused, the authors need to rewrite it and make it more clearly.

13.  Line 441, “compound 1 and 2” should be changed into two exact names of compound.

14.  Line 445, it is not significant activity or it is no activity because of the extract possessing an IC50>100 μg/mL.

15.  There are many mistakes in the format of reference citations. The authors need to uniform the formats according to the Guideline for Authors. Besides, the scientific name of species should be italic. 

Author Response

The detailed comments are:  

  1. Line 50, it is not correct to take Iran and China as south east countries.

-Response, Agreed, correction has been made in the revised manuscript.

  1. Line 56, “in vitro” should be italic. “as” in Line 57 should be “such as”.

-Response, Agreed, correction has been made in the revised manuscript.

  1. For the figures in the text, the wavy lines in Figure1 and Figure 2 should be removed; No species have been shown on the map of Turkey in Figure 3 while the figure title has mentioned Turkey; The exact number of compounds and references are need to attach on each histogram in Figure 4 and Figure 5; The “Figure 5.1” should be “Figure 6”, and the structures and names of each compounds should be more clear and large for readable. The resolutions of Figure 1-3 are needed to improve.

-Response, Agreed, correction has been made in the revised manuscript.

  1. For the tables in the text, the colons in Table1 and dots behind numbers in Table 3 should be removed; Table 3 need to add a column named “Chemical classification” and all compounds need to reorganize following the chemical classification.

Response, Agreed, correction has been made in the revised manuscript.

  1. Line122-123, the sentence is confused, underground parts should not belong to plant organs.

Response, Agreed, correction has been made in the revised manuscript.

  1. Line 142, the citation “Badruddeen et al.,2012; Ved, et al., 2016” is not consistent with citation format, it should be changed into “[41, 42]”. The same issue also can be found in Line 275.

Response, Agreed, correction has been made in the revised manuscript.

  1. Line 169-170, the classification of compounds is disordered. Many types in this sentence are overlapped, such as phenolic and flavonoids. The classification should be redone based on IUPAC rule or followed authoritative literatures and give the corresponding citation.

Response, Agreed, correction has been made in the revised manuscript.

  1. The scientific name of species should be used the formats of italic and full name for the first appear. “O. sericea” in Line 276 and “O. stenoloba” in Line 280 need to replace to “O. sericea” and “O. stenoloba”; “P. fastigiataand F. oxysporum” in Line 284 should be changed into “Phialophare fastigiata and Fusarium oxysporum”.

Response, Agreed, correction has been made in the revised manuscript.

  1. Line 328, the Unit of bioactivity values should be added. The same issues also present in Line 345-347, Line 379, Line 485, etc.

Response, Agreed, correction has been made in the revised manuscript.

  1. Line 339-341, based on the given IC50values, the authors have made a wrong result for the activity comparation.

Response, Agreed, correction has been made in the revised manuscript.

  1. “mg/mL” or “mg/ml” should be only choice one and keep it in whole text.

Response, Agreed, correction has been made in the revised manuscript.

  1. Line 410-412, the sentence is confused, the authors need to rewrite it and make it more clearly.

The 50 µg/mL ethanolic extract from aerial parts of O. Sericeum exhibited significant cytotoxity activity against the breast cancer cancer cells (MCF-7) with significantly decreased cell viability (28.76±11.31 %) [92].

  1. Line 441, “compound 1 and 2” should be changed into two exact names of compound.

The isolated hispidone and (2S)-5,2-dihydroxy-7,5-dimethoxyflavanone.

  1. Line 445, it is not significant activity or it is no activity because of the extract possessing an IC50>100 μg/mL.

Response, there were some unit errors for the IC50 values.

  1. There are many mistakes in the format of reference citations. The authors need to uniform the formats according to the Guideline for Authors. Besides, the scientific name of species should be italic. 

Response, Agreed, correction has been made in the revised manuscript.

Reviewer 2 Report

Ethnobotanical, Phytochemistry, and Pharmacological activity of Onosma (Boraginaceae): An updated review by Ahmed Aj. Jabbar et al describs compounds reported along with biological activity of the plants of the genus Onosma. The content of the manuscript covers the distribution, chemical profiles and various pharmacological activities of the extracts and isolates in detail. This is a well-organized comprehensive review, providing plentiful knowledge for further utilization of this genus.

Minor revisions such as following:

1. P5L142: Please replace ‘(Badruddeen et al., 2012; Ved, et al., 2016)’ as citation numberings.

2. Table 2: ‘Turkish’   ‘Turkey’

3. P25L213: ‘50% (LD50) and 90% (LD90)’   ‘50% (LD50) and 90% (LD90)’

4. P26 L263: ‘E. Coli’   E. coli

5. P26 L280: ‘O. stenoloba’   O. stenoloba

6. P26 L296: Aspergilus flavus  A. flavus

7. Names ‘...Naphthoquinone (33), Flavonoids (30), Hydrocarbon (23), Phenolic (22), Ester (18), Alkaloids (17), Terpenoids (11), Carboxylic acid (10), Fatty acids (9), Hydroxycoumarin (8), Alcohol (5), Aromatics (5), Liganin (4)’ do not need to be capitalized unless they are at the beginning of a sentence.

8. Revisions were required for some references, such as the lack of the pages for Ref.s [1] & [3], and the wrong volume number for Ref. [24]. And it is better to translate the title of Ref. [94] into English.

Author Response

Minor revisions such as following:

  1. P5L142: Please replace ‘(Badruddeen et al., 2012; Ved, et al., 2016)’ as citation numberings.

Response. The citation has been numbered.

  1. Table 2:‘Turkish’  →  ‘Turkey’

Response, Done.

  1. P25L213:‘50% (LD50) and 90% (LD90)’  →  ‘50% (LD50) and 90% (LD90)’

Response, corrected accordingly.

  1. P26 L263: ‘E. Coli’  →  ‘ coli

Response, corrected accordingly.

  1. P26 L280:‘O. stenoloba’  →  ‘ stenoloba

Response, corrected accordingly.

  1. P26 L296:‘Aspergilus flavus’  →  ‘ flavus

Response, corrected accordingly.

  1. Names ‘...Naphthoquinone (33), Flavonoids (30), Hydrocarbon (23), Phenolic (22), Ester (18), Alkaloids (17), Terpenoids (11), Carboxylic acid (10), Fatty acids (9), Hydroxycoumarin (8), Alcohol (5), Aromatics (5), Liganin (4)’ do not need to be capitalized unless they are at the beginning of a sentence.

Response, Agreed and has been corrected.

  1. Revisions were required for some references, such as the lack of the pages for Ref.s [1] & [3], and the wrong volume number for Ref. [24]. And it is better to translate the title of Ref. [94] into English.

-All the required information regarding the reference were added in the revised manuscript.

And reference 94 replaced with its English version.

Reviewer 3 Report

The manuscript is a detailed updated review that provides valid information and references on literature devoted to investigation of natural compounds isolated from Onosma spp, and of their biological activities.

After a general introductive part on Onosma spp, geographical distribution and traditional uses, the authors concentrate on relevant papers devoted to toxicity studies and pharmacological activities, supported by a consistent and coherent list of references.

The review appears to be complementary to two previous reviews published in 2013 and 2021 (listed as ref. 23, 24) and provides further information and support to investigators involved in the same area of research.

Author Response

Comments and Suggestions for Authors

The manuscript is a detailed updated review that provides valid information and references on literature devoted to investigation of natural compounds isolated from Onosma spp, and of their biological activities.

- Response, totally agreed and we hope we covered all the necessary information need to be mentioned regarding the Onosma ssp.

After a general introductive part on Onosma spp, geographical distribution and traditional uses, the authors concentrate on relevant papers devoted to toxicity studies and pharmacological activities, supported by a consistent and coherent list of references.

-Response, you are right dear doctor, thanks for your comments.

The review appears to be complementary to two previous reviews published in 2013 and 2021 (listed as ref. 23, 24) and provides further information and support to investigators involved in the same area of research.

- Response, we are agreed with your comments, but our study provides more precise information regarding the Onosma ssp in a more accurate and clear way.

Round 2

Reviewer 1 Report

The version has revised most of my comments, and get much improvment. I agree to aceept it for the publication on molecules.